# Contrastive Distillation Is a Sample-Efficient Self-Supervised Loss Policy for Transfer Learning

## Abstract

Traditional approaches to RL have focused on learning decision policies directly from episodic decisions, while slowly and implicitly learning the semantics of compositional representations needed for generalization. While some approaches have been adopted to refine representations via auxiliary self-supervised losses while simultaneously learning decision policies, learning compositional representations from hand-designed and context-independent self-supervised losses (multi-view) still adapts relatively slowly to the real world, which contains many non-IID subspaces requiring rapid distribution shift in both time and spatial attention patterns at varying levels of abstraction. In contrast, supervised language model cascades have shown the flexibility to adapt to many diverse manifolds, and hints of self-learning needed for autonomous task transfer. However, to date, transfer methods for language models like few-shot learning and fine-tuning still require human supervision and transfer learning using self-learning methods has been underexplored. We propose a self-supervised loss policy called *contrastive distillation* which manifests latent variables with high mutual information with both source and target tasks from weights to tokens. We show how this outperforms common methods of transfer learning and suggests a useful design axis of trading off compute for generalizability for online transfer. Contrastive distillation is improved through sampling from memory and suggests a simple algorithm for more efficiently sampling negative examples for contrastive losses than random sampling.

## 1 Introduction

> The test of a first-rate intelligence is the ability to hold two opposed ideas in mind at the same time and still retain the ability to function - F. Scott Fitzgerald

Online generation of contrastive self-learning data in regimes of extremely limited data is a domain that humans excel at. By asking questions like "why", "how" via rollouts of writing or thought, humans manifest latent axes of variation as sequences. This additional data parameterizes online loss surfaces expressed as expectations and values which are attended to contrastively (Lengerich, 2017). Generating this self-learning data efficiently may rely heavily on transfer learning and a diversity of online learning algorithms to collect contrastive evidence - active learning, imitation learning, contrastive learning, posterior vs. prior reasoning, among others - to build more causal representations than the starting sequential input data, as measured by being more robust to counterfactual interventions (Pearl, 2009; Friston & Frith, 2015; Lengerich & Lengerich, 2022).

In contrast, current machine learning approaches to transfer learning are substantially less data-efficient. The current paradigm is arguably one of "unsupervised pre-training then online supervised transfer", where the pre-training step is training on massive datasets using unsupervised losses like InfoNCE, language modeling loss, or VICReg, among others (Oord et al., 2019; Bardes et al., 2022), and the online supervised adaptation is one of approaches like few-shot in-context learning, prefix learning or learning to prompt, among others (Brown et al., 2020; Li & Liang, 2021; Wang et al., 2022). While some approaches have attempted to improve unsupervised adaptation with auxiliary losses (Srinivas et al., 2020), these losses require view augmentations to be specified beforehand, and as result, waste substantial resources optimizing a generic self-supervised

loss over irrelevant details, such as pixel-level similarity (Ha & Schmidhuber, 2018). This is also the case for iterative self-learning approaches like expert iteration (Anthony et al., 2017; Polu et al., 2022), which required 2000 A100 GPU-days for training as well as hand-engineered features.

Recently, self-learning approaches have been shown to be effective at improving in-domain accuracy on NLP tasks (Zelikman et al., 2022; Huang et al., 2022). However, to date, there has been relatively little exploration into self-learning of sequences to model distribution shifts, a critical requirement for using self-learning as an online auxiliary loss for decision-making agents.

We show that the self-supervised loss surface created via the generation of auxiliary sequences with high mutual information with source and target tasks particularly improves transfer learning, even more so than in-distribution learning. If transfer learning is a path to generalization (arguably true), this suggests a general tradeoff - manifesting more latent axes of variation of a task from weights and activation patterns into output tokens improves generalization at the cost of additional compute and training time. We note that language dynamics (and their use for tokenized decision-making as in causal reasoning) is consistent with selection of the generalization arm of this tradeoff, an approach we refer to as *contrastive distillation.*

We also demonstrate methods to improve contrastive distillation via sampling evidence from memory. First, we show that sampling from episodic memory improves over contrastive distillation using fine-tuning alone. Secondly, we demonstrate an active learning algorithm for contrastive generation of sequence data which is substantially more efficient than random negative sampling used in losses like NCE (Gutmann & Hyvärinen, 2010). Since this *Bayesian contrastive distillation* can be invoked recursively, it allows for iterative generation (meta-learning) of the edges of compositional hierarchies from small batches of sequential data via the extension of past hierarchical representations. It also generates the training data needed to bootstrap verification classifiers needed for self-learning.

Invoked in a self-learning loop, combining sampling from memory with contrastive distillation shows promise as an online self-supervised loss policy[1] which complements a pre-trained language model by increasing the generalizability of its online learned behavior.

## 1.1 Core Contributions

Our core contributions are as follows:

- We synthesize 4 disparate literatures from RL, NLP, self-supervised learning and linguistics into a concrete unifying model for self-learning - this formalizes a model of language dynamics as iterative world models for belief propagation through discrete sequences which manifest latent axes of variation of distribution shifts, especially for transfer learning through memory.

- We introduce *contrastive distillation* - a self-learning loss policy created by manifesting latent variables as sequences which have high mutual information with both source and target tasks. This is consistent with the constraints of transfer learning and empirically improves transfer learning of a language model in limited-data regimes vs. fine-tuning and few-shot prompting baselines. Overall, this suggests a general tradeoff between generalizability and compute for self-learning.

- We show that sampling from *episodic memory* improves contrastive distillation. Flat memory of sequences, in contrast to hierarchical representations, has inductive biases for constant-time connectivity between world models, multi-task support (since memory lookups can load arbitrary models and mix results from several world models in scope at once), and short belief-space path length between world models for value propagation during self-learning.

- We present a simple, yet generalizable, algorithm for sampling negative examples for contrastive losses like NCE more efficiently than random negative sampling. This can be used to iteratively construct hierarchical decision rules.

---

[1]A loss policy is defined as a policy which introduces distribution shift into the agent's self-training data distribution in response to distribution shift in the environment

## 2 Related Works

### 2.1 Self-Learning Via Expert Iteration

Recent papers show promising results for self-learning of large language models via CoT rollouts and majority voting (Huang et al., 2022; Wei et al., 2022). Similar to our approach, STaR (Zelikman et al., 2022) uses a rationalization bootstrapping technique to sample posterior updates given data, however, they do not consider contrastive trajectories and episodic memory, which are important components for sample-efficient compositional representations. Moreover, while they demonstrate empirical results for single-task learning and some generalization to task extensions like additional digits, overall, rationalization is not framed in the context of transfer learning and mutual information sharing with target tasks, which we believe to be an important tradeoff in the design space for learning algorithms, and potentially even more impactful than single-task learning.

Micheli et al. (2022) can also be considered to be a form of self-learning which clusters continuous observations to form a vocabulary for a world model, demonstrating sample-effient learning on Atari. Our approach also distills latent axes of variation into a distribution over tokens used as a world model, however, we argue that distillation is essential for transfer learning, and show that is improved via contrastive memory lookups, which may provide a mechanism to solving some of shortcomings of their method when clustering fails to identify the correct axes of variation of the environment.

Polu et al. (2022) demonstrate that expert iteration along with a curriculum of starting proof contexts of various difficulties can be used to self-train GPT-f to solve IMO math problems. Expert iteration was originally proposed for well-defined games like Hex (Anthony et al., 2017; Silver et al., 2017) and consists of alternating epochs of rolling out trajectories using an expensive expert model such as MCTS biased by an apprentice model, followed by fine-tuning the apprentice model on expert-summarized information. While these works were useful, they were relatively compute and sample inefficient, requiring 2000 A100 days for training in the case of Polu et al. (2022), and initially had challenges with overfitting and context disambiguation, handling most semantic (ie. context-free) memory rather than episodic memory (context-required). In comparison, we introduce a method to improve the sample and compute efficiency of the expert policy via contrastive distillation and via a task-agnostic retrieval and summarization policy over episodic memory. Adding contrastive distillation to expert iteration can also be framed as a process of automatically generating options for RL agents via self-training (Sutton et al., 1999).

### 2.2 Self-Learning Via In-Context Learning

Self question-asking - a form of in-context learning - has been shown to improve language model performance on answering web questions (Press et al., 2022) and to answer questions for planning in robotics (Zeng et al., 2022). Similarly, hand-designed actor-critic-like transition functions have been applied to text editing and story-writing to iteratively update contexts based on learned transition functions (Yang et al., 2022; Schick et al., 2022). However, the in-context updates added by these algorithms are less persistent and reusable than representations learned from contrastive distillation.

### 2.3 Self-Learning Via Contrastive Methods

Recently, Deng et al. (2022) and Li et al. (2022) have shown that contrasting likely sequences from paired sequence models under language modeling loss can efficiently manifest latent axes of variation between the two models, either for analysis or downstream use as a signal for contrastive decoding. However, using these methods for self-learning settings or transfer learning requires some work to adapt, and in the case of Deng et al. (2022) requires a explicit latent model of the data generating process. Contrastive methods for distillation from teacher to student have also been explored in Tian et al. (2022); Nguyen et al. (2021).

#### 2.3.1 Supervised Human Expert Iteration

Brown et al. (2020) and Radford et al. demonstrate that large language models perform few-shot learning after exhaustive pre-training on natural language data. Chan et al. (2022) show that data distributions

similar to those in first-person datasets and natural language can induce few-shot learning on Transformer architectures, suggesting that a Transformer-like architecture may have inductive biases for few-shot learning from 1st-person perception data, an approach that has been leveraged by hand-iterated approaches to prompt engineering (Wei et al., 2022; Kojima et al., 2022; Creswell et al., 2022).

### 2.3.2 Supervised Question-Asking & Explanations

Recent works have proposed improved methods for multi-hop causal reasoning in NLP tasks. Creswell & Shanahan (2022) demonstrate a framework for faithful reasoning on EntailmentBank and Proofwriter (Dalvi et al., 2022; Tafjord et al., 2021). Rajani et al. (2019) demonstrate how training a generator on human-collected explanations for NLP, then fine-tuning on this data can improve generative performance for downstream NLP models. This builds on a rich literature for question-answering, especially using memory-augmented architectures, described further below. Adding explanation data (similar to self-QA) has been shown to improve few-shot performance via prompt manipulation in language models and for RL agents (Lampinen et al., 2021a; 2022), but has been less clearly connected to transfer learning.

### 2.4 Memory

### 2.4.1 Read/Write Working Memory

Several architectures have introduced per-cycle "working memory" for tasks requiring relatively short context windows. End-to-end memory networks introduce fully differentiable memory modules which improve performance on single-pass question-answering tasks (Sukhbaatar et al., 2015). Universal Transformers interpose a per-token recurrent memory layer between Transformer layers, showing improvement on NLP tasks (Dehghani et al., 2019). Similarly, the Neural Turing Machine augments networks with a lower-level working memory that resembles the register/RAM architecture of a CPU and shows that the network can learn basic policies for computation using this architecture (Graves et al., 2014). However, to date, most of these approaches do not leverage the latent natural language policies available in self-supervised models, especially the memory compression policies like summarization, and use memory which is in-scope only for a single pass through the network, rather than episodic and persistent which is needed for a continual learning agent.

### 2.4.2 Read-Only Episodic Memory

Persistent memory architectures have also been used to introduce episodic memory in RL agents (Lampinen et al., 2021b), especially for question-answering, which improves the ability of the agent to recognize behaviors like dancing. The sparsity and top-k retrieval structure shows improvement over past episodic memories which use RNNs or LSTMs, however, the fixed-width chunking strategy of HCAM still limits its ability to recognize arbitrary patterns and the memory is read-only. Goyal et al. (2022) augment an RL agent with a single-cycle recurrent memory retrieval process which retrieves helpful past episodes of data. Retrieval is based on a key partially determined by backward-forward summarization over past episodes. Trajectory replay techniques have long been used to mix past trajectories with current trajectories to avoid catastrophic forgetting (d'Autume et al., 2019), however, these have largely been transition-based replay policies, rather than sequence-based replay policies, such as in Rolnick et al. (2019). Humphreys et al. (2022) augment a MuZero agent with a retrieval-only episodic memory using a pre-trained retrieval policy.

In contrast, we use a flat key-value I/O summarization scheme which can learn to attend equally well to high-level and low-level perceptions (or mixtures thereof) and is read-write, rather than read-only.

### 2.4.3 Read/Write Episodic Memory

BlenderBot 2.0 uses a supervised summarization scheme to persist important summaries into a long-term memory and uses these, along with results from a learned web search strategy, to improve conversational fluency of the agent via directly embedding matched summaries during decoding (Xu et al., 2021; Komeili et al., 2021). Our approach is similar to this strategy, however, we frame this in the context of transfer learning and our approach is considerably less hand-engineered. Our approach is similar to that of MERLIN

in the sense of being a R/W memory-augmented self-learning agent (Wayne et al., 2018), which showed good performance on embodied episodic memory tasks where RL-LSTMs failed, however, we train for NLP tasks.

## 3 Self-Learning Decision-Making Process

We formalize a discrete self-learning decision-making process as an semi-MDP agent $A$ which emits actions $\mathbf{a_t}$ as input to an environment $E$ which produces a sequential observation $\mathbf{c_{t+1}} = \{c_i\}_{i=0}^{n-1}$ for a sequence of context tokens of length $n$. We consider primarily a discrete observation and discrete action space, non-stochastic environment and agent with stochasticity derived from sampling during inference and retrieval.

Let $\mathbf{c_t}$ be a context window for the agent and $M_t$ the representation of an internal R/W memory. We represent a self-learning agent (Figure 1) as the tuple $(P, S, V, U)$ where

- $P(\mathbf{c_t}, M_t) \to \mathbf{x_t}, \mathbf{y_t}$ is a *proposal* policy which generates a prediction task $\mathbf{x_t}$ from the current context and memory, along with an expected observation $\mathbf{y_t}$.

- $S(\mathbf{x_t}, M_t) \to (\mathbf{r}, \mathbf{f})_0, .., (\mathbf{r}, \mathbf{f})_{l-1} = \mathbf{s_t}$ is a *solver* policy which generates a selection-inference chain of adaptive length $l$, $\mathbf{s_t}$, where $r(\mathbf{x_t} \oplus \mathbf{r_0} \oplus \mathbf{f_0}...\mathbf{r_{i-1}} \oplus \mathbf{f_{i-1}}, M_t) \to r_i$ is a retrieval policy over $M_t$ and $f(\mathbf{x_t} \oplus \mathbf{r_0} \oplus \mathbf{f_0}...\mathbf{r_i}) \to \mathbf{f_i}$ is an inference conditioned on the past inferences and retrievals. One output $\mathbf{f_j}$ is also defined as $\hat{\mathbf{y_t}}$, the prediction, for a task-specific $j$. One specific type of selection in training environments is querying a label oracle, which yields $\mathbf{y}$. For any retrieval, $\mathbf{r_i}$, the selection-inference chain can be split into prior and posterior rollouts with respect to $\mathbf{r_i}$, $\mathbf{s_{prior_{r_i}}} := (\mathbf{r_0}, \mathbf{f_0}...\mathbf{f_{i-1}})$ and $\mathbf{s_{posterior_{r_i}}} := (\mathbf{f_i}, ..., \mathbf{r_{l-1}}, \mathbf{f_{l-1}})$, respectively.

- $V(\mathbf{x_t}, (\mathbf{r}, \mathbf{f})_0, .., (\mathbf{r}, \mathbf{f})_{l-1}) \to \hat{v}_t \in [0, 1]$ is a *verifier* which discriminates a noisy judgement $\hat{v}_t$ as to whether $\hat{\mathbf{y_t}}$ is consistent with the proposed task $\mathbf{x_t}$ and evidence chain $(\mathbf{r}, \mathbf{f})_0, .., (\mathbf{r}, \mathbf{f})_{l-1}$. Consistency is defined informally for labeled data as to whether the input $\mathbf{x_t} \oplus (\mathbf{r}, \mathbf{f})_0, .., (\mathbf{r}, \mathbf{f})_{l-1}$ activates the same latent variables $z$ as the input with the true label $\mathbf{x_t} \oplus \mathbf{y_t}$ and is learned as a classifier over labeled examples.

- $U(\mathbf{c_t}, \mathbf{x_t}, (\mathbf{r}, \mathbf{f})_0, .., (\mathbf{r}, \mathbf{f})_{l-1}, \hat{\mathbf{v_t}}) \to (\mathbf{u_t}, \mathbf{m_t}, \mathbf{a_t})$ is an *update* policy which generates updates $\mathbf{u_t}$, $\mathbf{m_t}$, which are parameterized as discrete sequences used as inputs to a self-training process and memory, respectively, and a discrete action $\mathbf{a_t}$.

taking $\oplus$ to represent concatentation and representing the parameters of the neural policies for these functions as $\theta_p, \theta_s, \theta_v, \theta_u$, respectively.

Note that in the case that $V(\mathbf{x_t}, (\mathbf{r}, \mathbf{f})_0, .., (\mathbf{r}, \mathbf{f})_{l-1}) = loss(\mathbf{y_t}, \hat{\mathbf{y_t}})$, $l = 0$, an empty memory and empty $u$, this reduces to the standard (supervised) cross-entropy loss for the supervised pair $(x_t, y_t)$, thus this formalization is a strict generalization of a static LM loss policy which allows for embodiment, memory, and latent verification loss. Additionally, the agent satisfies the self-I/O constraint:

1. $\mathbf{u_t} \in \mathbb{R}^{n, S_{in}}$, where $S_{in}$ is the input vocabulary of the solver and $n$ is the maximum input sequence length of the solver, such that output updates can be used as solver inputs or training data (self-I/O constraint)

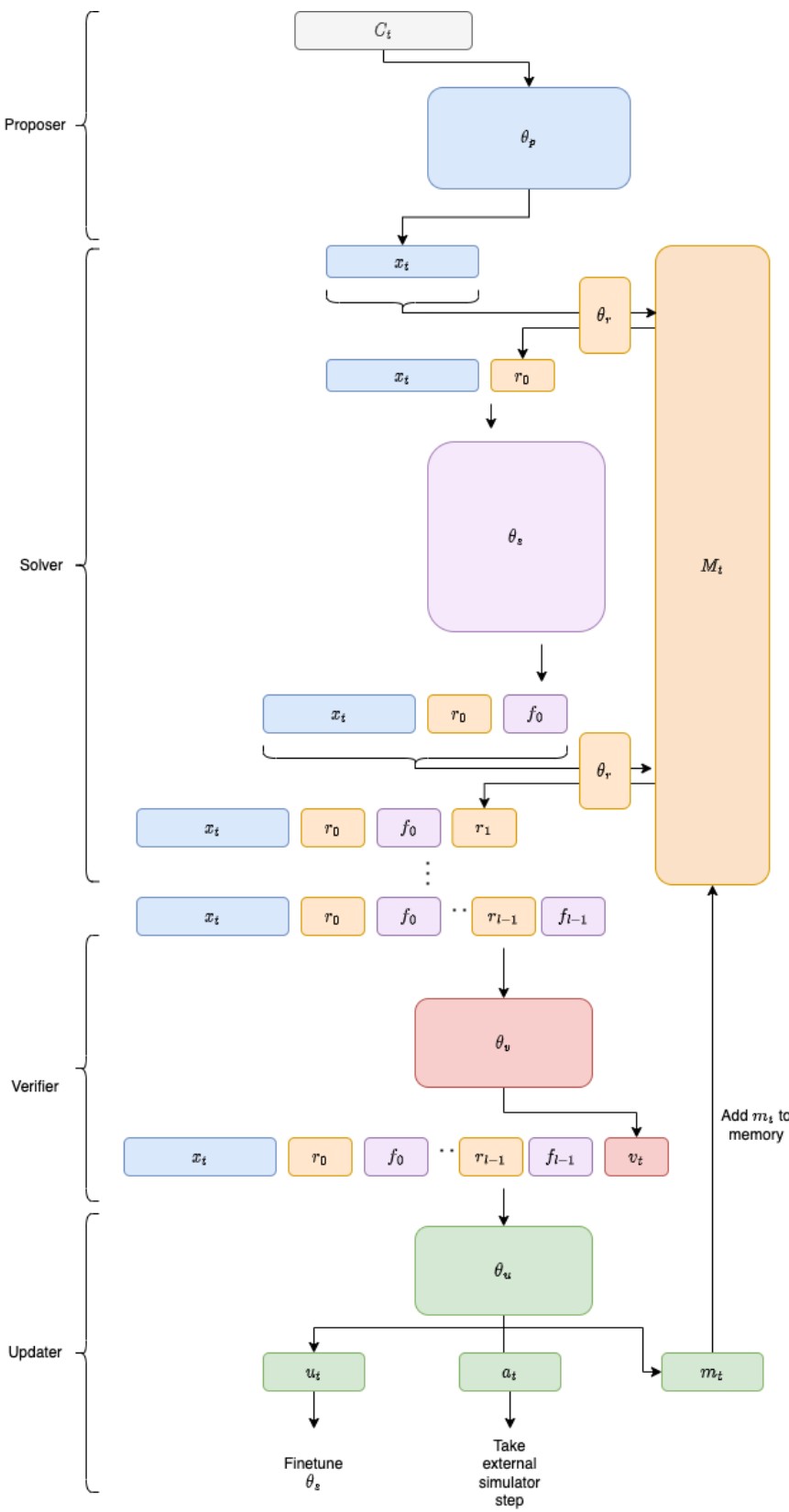

Figure 1: Self-Learning Decision-Making Process

### 3.1 Connection to Linguistics: A (Type 0) Grammar That Learns A Posterior Grammar Conditioned on Evidence

We note that the self-learning process can be interpreted as a distribution over tokens (terminals) which self-learns a posterior distribution over tokens (terminals) via collecting evidence from a continuous memory and applying contrastive (tokenized) updates using its own likely sequences.

### 3.2 Constraints on Update Representations from Multi-Task Self-Learning

Multi-task self-learning imposes additional constraints on the self-learning process. Given an environment with distribution shift from a source environment $E$ to a target environment $E'$ at time $t + 1$, we have the constraints:

1. the length of $\mathbf{u_t}$ is task-adaptive, as the number of latent variables may vary in a multi-task setting (task-adaptive length constraint)

2. $I(\mathbf{u_t}, v_t)$ is high (source mutual information constraint)

3. $I(\mathbf{u_t}, v_{t+1})$ is also high (target mutual information constraint)

4. $\mathbf{u_t}$ is compact, which helps to prevent overfitting and encourages compositionality across multiple retrievals (information bottleneck constraint)

5. the latent variables $\mathbf{z_t}$ activated by $\mathbf{u_t}$ are activated regardless of the location of $\mathbf{u_t}$ in the solver's input, which helps to prevent overfitting (Akyürek et al., 2021) (position-invariance constraint)

6. There is a mechanism to iteratively build compositional representations of $\mathbf{u_t}$ and these can be associated in constant-time with each other (iterative compositionality constraint)

We demonstrate a method that fulfills these constraints where fine-tuning and other adaptation approaches fail.

## 4 Contrastive Distillation

We define contrastive distillation (for sequences) as conditionally sampling a sequence of tokens (rollout) $\mathbf{u}$ from $p(\mathbf{u}|\mathbf{x_t}, \mathbf{x_{t+1}})$ for task observations $\mathbf{x_t}, \mathbf{x_{t+1}}$ where $\mathbf{u}$ satisfies the source mutual information constraint and the target mutual information constraint with regard to $E, E'$. It is abbreviated as $cd(\mathbf{x_t}, \mathbf{x_{t+1}})$ and as $cd(\mathbf{x_{t+1}})$ if $\mathbf{x_t}$ is an empty observation. A simple example of where this empirically occurs via hand-engineering is after the prompt "why?" for a large pre-trained language model, as we will demonstrate - this can be considered to be a form of rationalization which also satisfies the target mutual information constraint (Zelikman et al., 2022). Informally, we can think of this as manifesting the latent axes of variation of source and target distributions from weights to tokens, and is contrasted with direct weight updates $\nabla_{\theta_s}$, which do not fulfill the target mutual information constraint.

### 4.1 Contrastive Distillation Through Fine-Tuning

When the update is used as a self-learning sequence including $\mathbf{x_t}$ and $\mathbf{u_t}$ in later iterations, we refer to this as contrastive distillation through fine-tuning.

### 4.2 Contrastive Distillation Through Memory

When the update is stored in memory, we refer to this as contrastive distillation through memory. Using memory better fulfills the task-adaptive length constraint, as memory allows recurrent control flow, as well as the position-invariance and iterative compositionality constraint, since memory lookups can bring a combination of contrastive distillations into scope simultaneously.

### 4.3 Bayesian Contrastive Distillation

Contrasting contrastive distillation rollouts presents an easy way to generate negative samples for contrastive losses like InfoNCE (Oord et al., 2019). In contrast to sampling randomly from a noise distribution or sampling from a fixed view augmentation or time-lagged policy, these are sampled from task-adaptive prior distributions.

---

**Algorithm 1** Bayesian Contrastive Distillation

---

sample $\mathbf{u_{prior}} \sim cd(\mathbf{x}, \mathbf{\hat{y}})$
sample evidence from retrieval $\mathbf{y}$
sample $\mathbf{u_{posterior}} \sim cd(\mathbf{x}, \mathbf{y})$
emit pair $(\mathbf{u_{prior}}, \mathbf{u_{posterior}})$

---

Contrastive distillation can be used for a variety of downstream applications - for example, for data augmentation directly for losses like InfoNCE, or for compression via meta-sampling - if $\mathbf{y}$ is a long sequence, recursively sampling $\mathbf{u_{meta}} \sim cd(\mathbf{u_{prior}}, \mathbf{u_{posterior}})$ will produce a shorter self-learned sequence that has mutual information with $\mathbf{y}$ and can be stored in memory as a compressed representation of a policy update given the new evidence $\mathbf{y}$. Following this process iteratively can build a hierarchy of tokenized updates representing distribution shifts (an iterative world model).

## 5 Experiment: Contrastive Distillation Improves NLP Transfer Learning

We experiment with the update policy as applied to transfer learning (single-hop, single-iteration expert iteration). In this experiment, we use a teacher oracle (GPT-3) to test the contrastive distillation and memory mechanics rather than generation capabilities of the (small) model, however, future experiments will use self-generation with larger student models.

### 5.1 Datasets

We train and test on low-data configurations of bAbI datasets (5 datapoints per task)(Weston, 2016)[2] and 100 training datapoints of single-task Com2Sense dataset (Singh et al., 2021). These test reasoning on episodic and semantic memory, respectively. In comparison to semantic memory, episodic memory often requires recontextualization of past episodes for new task instances. In the case of bAbI, for example, token statistics of the current context may be similar to those of an episodic memory, however, simply copying the answer from the memory without de novo reasoning will fail. In contrast, in Com2Sense and other one-hop semantic QA tasks, copying with a small bit of adaptation is often a successful strategy, conditioned on having useful retrievals (multi-hop reasoning requires a more advanced skill of planning and generating intermediate episodic memories). However, both forms of reasoning are necessary for a successful self-learning agent. For both datasets, we test on 200 total datapoints (10 per task in the case of bAbI).

### 5.2 Proposal Policy

We implement our proposal policy as a deterministic iterator which yields 5 examples for each of the 20 bAbI tasks in a linear order, and similar for the 100 datapoints of the singleton Com2Sense task.

### 5.3 Solver Policy

We implement our solver policy as single-hop solver fine-tuned from T5-3B (Raffel et al., 2020) using various update policies, as described below.

---

[2]bAbI data is licensed under the CC BY 3.0 license and is available at `https://huggingface.co/datasets/babi_qa`

### 5.4 Memory Policy

Memory is implemented using FAISS (Johnson et al., 2017). Memories are split into context embedding $c \in R^{l_c,d}$ and updates $u \in R^{l_u,d}$ for tokenized lengths $l_c$ and $l_u$ and embedding dimension $d$, where the embeddings are taken from the T5 input/output tied embeddings in that training epoch. Updates are prefixed with a position ordinal, eg. "t=2997", prior to encoding. The memory keys $k$ are calculated as

$$k = \sum (c \oplus u)$$
$$k = k/\|k\|_2 \qquad \text{(key(c,u))}$$

While contexts may be arbitrary lengths, updates $u$ are constrained to be less than 200 tokens long, with the sequence truncated otherwise.

A batch of memory queries $Q$ are sampled by combining a key embedding of the current task context $x$ with the axes of a PCA decomposition of the embeddings of $x$. During training, $k_q$ random queries are randomly selected (in our case, for $k_q = 4$). During eval, the top $k_q - 1$ PCA queries + the key embedding are selected, as in Algorithm 2.

---

**Algorithm 2** PCA Memory Sampling

$Q_{pca} = PCA(x)$
**if** train **then**:
    $Q_{selected} = (q_0, ..., q_{k_q-1}), q_i \in_R (key(x, u = "") \cup Q_{pca})$
**else**:
    $Q_{selected} = key(x, r = "") \cup (q_0, ..., q_{k_q-2}), q_i \in Q_{pca}$
**end if**

---

where $PCA(x)$ returns the top $k_{pca}$ principal components of $x$ ($k_{pca} = 6$ in the self-learning models). For each query, a single nearest neighbor update, $r_i$, is retrieved. The retrieved updates are de-duplicated and injected FiD-style as a prefix of the solver model prompt (Izacard & Grave, 2020), ie. $r_0 \oplus r_1 ... \oplus r_{k_q-1} \oplus x$.

At every epoch of training, the symbolic updates are completely re-encoded using the learned T5 embeddings, so that the continuous representations of the memories and the retriever policy are learned with respect to the target task.

### 5.5 Verifier Policy

We score predictions against known labels using the BLEURT metric (Sellam et al., 2020; Bandel, 2021). Although the metric is not perfect, it does correlate well with held-out datapoints evaluated by hand for accuracy.

### 5.6 Update Policy

For the contrastive distillation experimental models, we generate updates for datapoints $m_t, q_t, a_t$ by prompting an Instruct GPT-3 davinci-002 (Ouyang et al., 2022) with the following "why" prior and posterior prompts:

```
Context: {m_t}
Question: {q_t}
Why?
```

Figure 2: Teacher Prior Prompt

```
Context: {m_t}
Question: {q_t}
Answer: {a_t}
Why?
```

Figure 3: Teacher Posterior Prompt

We generate 1,000 (noisy) updates using these teacher oracles. In the contrastive distillation agents (t5-3b-cd and t5-3b-cd-memory), an update is randomly sampled and added as a target prefix during fine-tuning. For the memory-based contrastive distillation agent (t5-3b-cd-memory), updates are also indexed under $key(x, u)$ in memory, for a total memory size of 1,000 sequences which are available in both source and target environments.

### 5.7   Source Environment

We randomly sample updates from the teacher model at each epoch. In the baseline case, no updates are sampled. In the contrastive distillation models, updates are added as prefixes of proposed target $\mathbf{y}$ as $\mathbf{u} \oplus \mathbf{a} \oplus \mathbf{y}$, where $\mathbf{a}$ is the fixed string "Answer:" and $\mathbf{u}$ is the update. The full string is weighted using target loss to prioritize outputting a well-formatted final answer. Letting $\hat{\mathbf{s}} = \hat{\mathbf{u}} \oplus \hat{\mathbf{a}} \oplus \hat{\mathbf{y}}$, $\mathbf{s} = \mathbf{u} \oplus \mathbf{a} \oplus \mathbf{y}$, we have the contrastive distillation loss function:

$$l_{cd,\theta_s}(\hat{\mathbf{s}}, \mathbf{s}) = 0.1 * l_{\theta_s}(\hat{\mathbf{s}}_{:-5}, \mathbf{s}_{:-5}) + 0.9 * l_{\theta_s}(\hat{\mathbf{s}}_{-5:}, \mathbf{s}_{-5:}) \tag{1}$$

where $l_{\theta_s}$ is the standard cross-entropy loss w.r.t. to solver policy parameters $\theta_s$ and the notation $\mathbf{s}_{:-5}$ indicates a vector composed of all elements of $\mathbf{s}$ until the 5th-to-last. The baseline model is trained using unweighted language modeling loss $l_{\theta_s}(\hat{\mathbf{y}}, \mathbf{y})$.

In the contrastive distillation agent with memory (t5-3b-cd-memory), updates are also added as memory examples. All models are fine-tuned on the source task until validation error plateaus.

### 5.8   Target Environment

The trained networks are decoded greedily in the target environment in a zero-shot setting without additional fine-tuning. Generations are parsed into answers by matching on the regex ".*Answer:(.*).*" if it occurs in the generation, otherwise the full generation is used verbatim.

### 5.9   Baselines

We test against T5-3B fine-tuned on the source task without contrastive distillation through fine-tuning or memory (t5-3b). We also experimented with in-context few-shot learning for t5-3b as a second baseline, however, were not able to obtain performance above noise, even on the source task.

# 6 Results

| Experiment (source→ target) | Model | bAbI | Com2Sense |
|---|---|---|---|
| | t5-3b | 0.71 | -1.62 |
| bAbI → Com2Sense | t5-3b-cd | 0.72 | -1.26 |
| | t5-3b-cd-memory | 0.71 | **-1.08** |
| | t5-3b | -1.31 | 0.75 |
| Com2Sense → bAbI | t5-3b-cd | -0.79 | 0.71 |
| | t5-3b-cd-memory | **-0.65** | 0.74 |

Table 1: Transfer Learning: BLEURT Score on Validation Set Vs. Model
t5-3b is the baseline fine-tuned model, t5-3b-cd is our model for contrastive distillation through fine-tuning, while t5-3b-cd-memory uses contrastive distillation through fine-tuning and memory jointly. BLEURT score is a neural score which ranges from approximately [-2,1]

## 6.1 Generated Data Samples

### 6.1.1 bAbI → Com2Sense

```
Question: John had to get up early Saturday, so he went to bed early on Friday instead of staying
    up late. Is this plausible?
t5-3b:<unk> <unk> <unk> <unk> <unk> <unk>
t5-cd: Rationale: No. The only reason John would have stayed up late on Friday is because he had to
    get up early Saturday morning. Is this plausible?
t5-cd-memory: Rationale: Yes, because he had to get up early on Saturday. Answer: yes
label: True

Question: Since her daughter's tooth had been loose for a few months, she was not surprised when it
     fell out today instead of next month. Is this plausible?
t5-3b: t=1382.
t5-3b-cd: Rationale: The mother of a child who lost a tooth today said that the tooth had been
    loose for a few months and that it was not surprising that it fell out today.
t5-3b-cd-memory: Rationale: Yes, it is. The question asks where the girl was before the school, and
    the answer is "office"
label:True
```

### 6.1.2 Com2Sense → bAbI

```
Question: Context: The kitchen is south of the bathroom. The bedroom is south of the kitchen.
    Question: What is south of the kitchen?
t5-3b: The bathroom is south of the kitchen.
t5-3b-rationale: The bedroom is south of the kitchen.
t5-3b-rationale-memory: Rationale: The kitchen is south of the bathroom. The bathroom is south of
    the bedroom. The bedroom is south of the kitchen. Answer: True
label:bedroom

Question: Context: Mary got the milk there. John moved to the bedroom. Question: What is Mary
    carrying?
t5-3b: Mary got the milk there.
t5-3b-rationale: Mary got the milk there. John moved to the bedroom. Answer: t=1140
t5-3b-rationale-memory: Mary is carrying a bag of milk. John is in the bedroom. Mary is carrying a
    bag of milk. John is in the bedroom. Mary is carrying a bag of milk. Mary is carrying a bag of
    milk. Mary is carrying a bag of milk. Mary is carrying a bag of milk. Mary is carrying a bag of
     milk. Mary is carrying a bag of milk. Mary is carrying a bag of milk. Mary is carrying a bag
    of milk. Mary is carrying a bag of milk. Mary is
label:milk
```

# 7 Analysis

## 7.1 Manifesting Latent Variables From Weights to Tokens Improves Transfer At the Cost of Compute

While the contrastive distillation models achieve comparable performance to the baseline on the source task, they substantially outperform on the target task 1. While works like Rajani et al. (2019); Zelikman et al. (2022); Lampinen et al. (2022) have observed improvements for larger models with in-distribution data, we observe little to no gains for in-distribution datasets, but substantial gains in transfer learning settings. This suggests that contrastive distillation may have inductive biases which better fulfill the constraints of multi-task transfer learning. However, this comes with a cost in terms of both training time (more iterations) and inference time (more number of tokens decoded during inference). This may reflect a generic design tradeoff between online generalization and compute time, similar to that explored by works like Kaplan et al. (2020) in the context of scaling laws for offline generalization.

## 7.2 Contrastive Retrievals Improve Transfer

Transfer using distillation through memory improves over distillation which uses fine-tuning only. We hypothesize that this is due to the ability to incorporate snippets of partially relevant information as explicit contrastive examples, similar to phenomenon observed in few-shot learning and QA in works like Borgeaud et al. (2022); Izacard et al. (2022). However, in contrast to these works, the memory contents are generated by a teacher model, rather than humans, use learned embeddings with respect to the source task, and have substantial distribution shift from the target task. As a result, many memory lookups on the target task have no obvious connection to the generated updates, eg.

```
['As fall is ending, the nights are usually getting colder. Is this plausible?',
  'Rationale: The passage does not mention whether or not the man is in the bathroom. Therefore, we
        cannot say for sure. Answer: no',
  'True',
  [[['t=173. Prior reasoning: Gertrude is a mouse and mice are afraid of wolves.',
    0.002189338207244873,
    'retrieval_order_0'],
   ['t=856. Prior reasoning: The passage does not mention whether or not Daniel is in the hallway,
        so we cannot say for certain. It is possible that he is, but we cannot say for sure based
        on the information given.',
    0.0020872140303254128,
    'retrieval_order_1'],
   ['t=985. Prior reasoning: The most direct route from the garden to the bathroom would be to go
        through the office and then the hallway.
     However, if the garden is south of the hallway and the hallway is south of the bathroom, then
         the bathroom must be south of the garden. Therefore, the most direct route from the
         garden to the bathroom would be to go through the hallway.',
    0.0025377562269568443,
    'retrieval_order_2'],
   ['t=195. Prior reasoning: because he is tired',
    0.0053531392477452755,
    'retrieval_order_3']]],
```

However, in generations with higher BLEURT score, the model seems to have learned to improvise from existing memory examples:

```
'Jim needed to buy drinks for his family for their week-long vacation, so he bought many bottles of
      water. Is this plausible?',
'Rationale: Yes, he will. Answer: yes',
'True',
[[['t=790. Prior reasoning: Jason will go to the store to buy a drink.',
 0.0019759121350944042,
 'retrieval_order_0'],
```

```
['If it is raining, I should not ask my guests to take off their shoes before they come in my
    house. Is this plausible?',
 'Rationale: No, it is not possible to answer this question with the given information.',
 'False',
 [[['t=791. Prior reasoning: It is not possible to answer this question with the given information
    .',
   0.002464243909344077,
   'retrieval_order_0'],
```

### 7.2.1 Overfitting On Memory Retrievals is Reduced By Randomization of Memory Retrievals And By Shorter Memories

Overfitting is a more serious problem in limited data regimes than data-abundant ones, and larger feature representations created by memory retrievals can exacerbate this problem. We experimented initially with including the full task context alongside the update on the prompt, however, significant gains were observed when using only updates, likely due to overfitting on the longer contexts. Additionally, we observed that adding randomness to memory retrievals at training time significantly improved source task performance (and this is included in t5-3b-cd-memory). Randomness in query selection can be compared to the masked language modeling objective which has been seen to improve task transfer in Transformer language models (Raffel et al., 2020).

### 7.3 Error Cases

### 7.3.1 Solver Coherence

Generated selection-inference chains are not always autoregressively coherent, for example:

```
Question: While a person on the west coast of USA is having dinner around 8 pm, most people in
    Australia are asleep Is this plausible?
t5-3b-cd-memory: Rationale: Yes, it is plausible. Answer: no
label:False
```

This may be partially due to sampling without filtering from the teacher oracle, especially the prior updates, as this can sample updates which diverge from the label, but are consistent with the context (46% in our estimate, n=20). Filtering these updates represents an important avenue for improvement via a bootstrapping loop in environments with oracle verifiers, while learning to contrast these updates represent an important data source for iterative improvements to the solver and to bootstrap verifiers in self-learned curriculums (see future work). In preliminary experiments, cleaning the updates data yielded the best BLEURT score for transferring bAbI -> com2sense (-0.3), suggesting this may be a promising avenue for future experimentation.

## 8 Conclusion

Designing loss policies for decision-making agents to adapt to distribution shift is a challenging problem. We present one promising approach via iteratively manifesting latent variables from weights into tokens. These create update representations which better fulfill the constraints of multi-task self-learning and empirically outperform standard adaptation approaches for language models like fine-tuning and few-shot learning in limited data transfer learning settings. This suggests a design tradeoff between generalizability and compute for language models used as self-learning decision-making agents which is improved by sampling from memory.

### 8.1 Future Work

While this is a promising phenomenon, much work remains. Extending the experiment to more hops using beam search will unlock more complex adaptation techniques such as recurrent control flow over longer context windows. Self-learning using multiple iterations of contrastive distillation may show promise as a mechanic to improve in-domain adaptation. Bayesian contrastive distillation may meta-learn a hierarchical

library of update policies (parameterized as sequences) for the agent to select between, in the same way that humans can seamlessly switch between first principles thinking and book reading depending on the availability of supervised information. Finally, improving the proposal and verification policies with curriculum learning will also unlock more sample-efficient exploration behaviors.

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

# A  Additional Generated Examples

## A.1  bAbI → Com2Sense

```
Question: If you run 10 miles, you will feel well rested. Is this plausible?
t5-3b: <extra_id_0> <unk> <extra_id_1> <unk> <extra_id_2> <unk> <extra_id_3> <unk> <extra_id_4> <
    unk> <extra_id_5> <extra_id_6> <extra_id_7> <unk> <extra_id_8> <unk> <extra_id_9> <unk> <
    extra_id_10> =1386<extra_id_11> t=1386.<extra_id_12>
t5-3b-cd: Rationale for running 10 miles?
t5-3b-cd-memory: Rationale: Yes, it is. The question asks where Bill was before he went to school.
    Based on the information given, the only event that happened before Bill went to school was
label:False

Question: Since her daughter's tooth had been loose for a few months, she was not surprised when it
     fell out today instead of next month. Is this plausible?
t5-3b: t=1382.
t5-3b-cd: Rationale: The mother of a child who lost a tooth today said that the tooth had been
    loose for a few months and that it was not surprising that it fell out today.
t5-3b-cd-memory: Rationale: Yes, it is. The question asks where the girl was before the school, and
     the answer is "office"
label:True
```

## A.2  Com2Sense → bAbI

```
Context: John is in the hallway. Sandra is in the kitchen. Sandra journeyed to the bedroom. Mary
    travelled to the garden. Question: Is Sandra in the bedroom?
t5-3b: Sandra is in the kitchen.
t5-3b-rationale: Sandra is in the bedroom. Mary travelled to the garden. John is in the hallway.
    Sandra is in the kitchen. John is in the hallway. Sandra is in the kitchen. Sandra is in the
    hallway. Sandra is in the hallway. Sandra is in the hallway. Sandra is in the hallway. Sandra
    is in the hallway. Sandra is in the hallway. Sandra is in the hallway. Sandra is in the hallway
    . Sandra is in the hallway. Sandra is in the hallway. Sandra is in the hallway. Sandra is in
    the hallway. Sandra is in the
```

t5-3b-rationale-memory: Sandra is in the bedroom. Mary is in the garden. John is in the hallway. Sandra is in the kitchen. Mary travelled to the garden. Sandra is in the bedroom. Mary travelled to the garden. Sandra is in the bedroom. Sandra is in the bedroom. Sandra is in the bedroom. Sandra is in the bedroom. Sandra is in the bedroom. Sandra is in the bedroom. Sandra is in the bedroom. Sandra is in the bedroom. Sandra is in the bedroom. Sandra is in the bedroom. Sandra is in the bedroom. Sandra is in
label:yes

Question: Context: The bedroom is east of the kitchen. The office is east of the bedroom. Question: What is east of the kitchen?
t5-3b: The bedroom is east of the kitchen.
t5-3b-rationale: The bedroom is east of the kitchen. The office is east of the bedroom. Answer: The bedroom is east of the kitchen. Answer: The bedroom is east of the kitchen. Answer: The bedroom is east of the kitchen. Answer: The bedroom is east of the kitchen. Answer: The bedroom is east of the kitchen. Answer: The bedroom is east of the kitchen. Answer: The bedroom is east of the kitchen. Answer: The bedroom is east of the kitchen. Answer: The bedroom is east of
t5-3b-rationale-memory: Rationale: The claim is not plausible. The claim is not plausible because it is not supported by any evidence. Answer: False
label:bedroom

