# OpenReview forum: "Contrastive Distillation Is a Sample-Efficient Self-Supervised Loss Policy for Transfer Learning"
_TMLR — Rejected by TMLR_

### Review · Reviewer_13Yt · 2023-02-20

**Summary Of Contributions:**

This paper addresses a subproblem of Natural Language Processing (NLP) transfer learning, in which low-data configurations are considered. To this end, the paper proposes a learning technique that conditionally samples training-efficient data from the given source and target dataset. However, the description of the proposed method is unclear, and the experiment setups seem to be problematic and do not fully verify the contributions. Therefore, I recommend rejecting this paper.

**Audience:**

Yes

**Claims And Evidence:**

No

**Requested Changes:**

- The authors should significantly improve their writing and organization. To be more specific, many terms and notations are not explained or defined at their first appearance. Descriptions of the model architecture in Figure 1 are not provided. Also, a comprehensive description of the core methods is lacking. The above weakness makes the paper hard to read and inconvincible.

- The authors should include more datasets and baselines and refine the experiment setups. It is more proper to test the effectiveness of transfer learning on newly learned student models instead of the teach oracle (GPT-3).

**Strengths And Weaknesses:**

## Paper strengths and contributions
**Motivation and intuition**
- The motivation of this paper, to transfer the ability of large-scale pre-trained models to new target domains with few data required, is practical.

**Technical contribution**
- The proposed method outperforms a simple fine-tuning baseline on two datasets (bAbI and Com2Sense).

## Paper weaknesses and questions

**Clarity**
- The presentation of this paper is not well-organized, and a lot of important information is missing. For instance, a comprehensive description of the three key methods in Sections 4.1 to 4.3 is lacking.
- Experimental details, including model architectures, the number of training iterations, and explanations of the experiment results, are missing.

**Oversell**
- I believe the authors oversell the core contributions. This paper only considers a specific few-shot NLP task, instead of general self-supervised transfer learning as described in the title and the abstract.


**Experiment setup**
- The experiments are not sufficient. Only one experiment result is presented, and no ablation study is included. Moreover, the proposed method is only evaluated on two datasets with 100 training data and 200 testing data, which is not sufficiently convincing even for the few-shot configurations.

- The experiment setup in Section 6 is suitable. While transfer learning aims to adapt models learned on a source domain to target domains, the authors utilized a large-scale pre-trained model and then fine-tuned it on an additional dataset and evaluated it on another. As the pre-trained model is already trained on other data, the experiment results in Table 1 do not directly represent the transfer ability between the two datasets.

- The paper only compares the proposed method against a simple fine-tuning baseline, which is insufficient. Domain adaptation techniques such as DANN [1] and ADAA [2] should be considered and discussed.

**Experiment details**
- Many details are missing, including the training iterations, learning rate, token length, the size of memories, etc.

**Experimental conclusions**
- Overall, the experiment of this paper is flawed and can not stand for the claimed contributions.

[1] Ganin, Y., Ustinova, E., Ajakan, H., Germain, P., Larochelle, H., Laviolette, F., ... & Lempitsky, V. (2016). Domain-adversarial training of neural networks. The journal of machine learning research, 17(1), 2096-2030.
[2] Tzeng, E., Hoffman, J., Saenko, K., & Darrell, T. (2017). Adversarial discriminative domain adaptation. In Proceedings of the IEEE conference on computer vision and pattern recognition (pp. 7167-7176).

---

> ### Author Response · Authors · 2023-02-20
> **Clarification?**
>
> Thanks for the review - will update with parameters and requested info.
>
> Would it be possible to clarify more on "As the pre-trained model is already trained on other data, the experiment results in Table 1 do not directly represent the transfer ability between the two datasets?" (as indicated in 5.9, there is a baseline for the few-shot T5 setting, but performance was not above noise; 0-shot T5 w/o finetuning or contrastive distillation evaluated on the target task is arguably a reasonable baseline for any easily accessible target information that the model had learned during pre-training)).

---

> > ### Comment · Reviewer_13Yt · 2023-02-21
> > **Re: Clarification?**
> >
> > Specifically, there are three datasets involved in the experiments, i.e., the pre-training dataset, bAbI, and Com2Sense. Therefore, when bAbI is the source dataset, the model learns from the pre-training dataset and bAbI, violating standard transfer learning setups.
> > Even though the authors argue that the model does not learn target information during pre-training, I feel it makes more sense to fix control variables by excluding the pre-training dataset.
> > Besides, taking the pre-training dataset as the source data and evaluating the model on bAbI or Com2Sense would also be a flexible experiment setup.

---

> > > ### Author Response · Authors · 2023-02-21
> > > **Follow-up**
> > >
> > > Thanks for the reply. This is the same setup as standard works like https://openreview.net/pdf?id=_3ELRdg2sgI and https://arxiv.org/pdf/2202.01344.pdf (pretrain, adapt to one task via fine-tuning, then test zero-shot generalization on a target out-of-distribution task), controlling for inductive biases carried from pretraining via the zero-shot baseline.

---

> > > > ### Comment · Reviewer_13Yt · 2023-02-23
> > > > **Re: Follow-up**
> > > >
> > > > Thanks for clarifying the experiment setups. I believe the few-shot learning baseline should be included and discussed to make the presented results more complete. The authors can refer to Table 1 in STaR (https://openreview.net/pdf?id=_3ELRdg2sgI) for possible baselines.

---

### Review · Reviewer_7N8x · 2023-03-17

**Summary Of Contributions:**

This paper proposes a self-supervised loss policy called contrastive distillation that improves transfer learning of a language model in limited-data regimes vs. fine-tuning and few-shot prompting baselines. The authors also claim that sampling evidence from episodic memory improves contrastive distillation. They also present an active learning algorithm for the contrastive generation of sequence data which is more efficient than random negative sampling used in losses like NCE. The authors show experimental results on bAbI datasets and Com2Sense datasets and report the BLEURT Score on the validation set.

**Audience:**

Yes

**Broader Impact Concerns:**

I do not believe this paper has major ethical concerns.

**Claims And Evidence:**

Yes

**Requested Changes:**

The proposed method is similar in its goal to multiple current methods that sought to improve performance in LLMs by using pre-trained knowledge from the memory, or few-shot domain transfer between datasets (see above). More experimental results are needed to establish the contributions of the proposed algorithm in comparison to previous works. Even if a comparison with fine-tuning-based methods is not performed, a comparison with some other in-context methods is desirable for proving the efficiency of this method.

**Strengths And Weaknesses:**

Strengths:
The paper tries to bring concepts from RL, NLP, self-supervised learning, and linguistics together for a few-shot domain transfer.

Weaknesses:
The organization of the paper could have been better to clearly explain the idea in the paper. However, the major weakness of the paper is the lack of comparison to various few-shot methods for improving prompt tuning [1, 2]. There are various other methods that improve few-shot learning using semi-supervised methods [3,4] and other methods that perform domain transfer in a low data regime [5,6]. This paper does not perform a rigorous experimental study on how the proposed method is different from the above existing work in terms of empirical performance.

[1] Vu, Tu, Brian Lester, Noah Constant, Rami Al-Rfou, and Daniel Cer. "Spot: Better frozen model adaptation through soft prompt transfer." Association for Computational Linguistics, ACL 2022

[2] Gu, Yuxian, Xu Han, Zhiyuan Liu, and Minlie Huang. "PPT: Pre-trained Prompt Tuning for Few-shot Learning." In Proceedings of the 60th Annual Meeting of the Association for Computational Linguistics, pp. 8410-8423. 2022.

[3] Zhou, Yuhang, Suraj Maharjan, and Beiye Liu. "Scalable Prompt Generation for Semi-supervised Learning with Language Models." arXiv preprint arXiv:2302.09236 (2023). (after submission)

[4] Lang, Hunter, Monica N. Agrawal, Yoon Kim, and David Sontag. "Co-training improves prompt-based learning for large language models." In International Conference on Machine Learning, pp. 11985-12003. PMLR, 2022.

[5] Guo, Xu, Boyang Li, and Han Yu. "Improving the Sample Efficiency of Prompt Tuning with Domain Adaptation." EMNLP 2022 (findings)

[6] Wang, Chengyu, Jianing Wang, Minghui Qiu, Jun Huang, and Ming Gao. "TransPrompt: Towards an automatic transferable prompting framework for few-shot text classification." In Proceedings of the 2021 Conference on Empirical Methods in Natural Language Processing, pp. 2792-2802. 2021.

---

### Review · Reviewer_o71a · 2023-03-26

**Summary Of Contributions:**

This paper claims to propose a self-supervised loss (contrastive distillation) which they claim improves transfer from a source to target domain. They perform their experiments on transfer between two reasoning tasks (bAbI and Com2Sense).
They show that contrastive distillation improves transfer learning compared to baselines.

**Audience:**

No

**Broader Impact Concerns:**

I do not think there are any concerns.

**Claims And Evidence:**

No

**Requested Changes:**

1. Change or add datasets to the paper. One example is the MRQA benchmark (and QA datasets in general) where transfer learning is a common thing to measure. The current two datasets are not sufficient to substantiate the point they are making.
1. Improve writing. There is a lot of vague writing and references (as mentioned in the previous section), and the writing in other sections can be simplified and improved to a large extent (sections 3 and 4 for example).

**Strengths And Weaknesses:**

**Strengths**
1. The related work is well covered and also well categorized.
1. The improvements in results (BLEURT score `-1.62` to `-1.08`) is reasonable (although the scores are in the negative).

** Weaknesses**
1. The paper is poorly written in many places. For example, there is no mention of how much the contrastive distillation improves over baselines either in abstract or introduction. The abstract is worded vaguely, for example, "learning compositional representations" implies that their method actually learns compositional representations, but there are no analyses regarding this. They claim that they "synthesize 4 disparate literatures ..", but that is an exaggeration in my opinion. They cover the related work well, but this is a methodology paper and I don't see related work as a contribution.
1. The improvements in performance are poor. The BLEURT score is in the negative. Although BLEURT score comparisons are relative and not absolute, given that bAbI -> bAbI performance is `0.72`, a bAbI -> Com2Sense score of `-1.26` is a huge drop. This shows that the transfer is bad.
1. The baseline is very weak. They use a single baseline to compare against, and from the outputs of the baseline (6.1), it seems extremely weak. Improving over that baseline, while not trivial, is not a metric of success in my opinion.
1. Their models seem to be using extra data generated by `Instruct-GPT`, whereas the baseline does not use it (implementation details are not clear). Hence, I don't think the comparison is fair.
1. The choice of datasets is arbitrary and very narrow. They only test on the low-data configurations of two datasets. Since they seem to be making comments about transfer learning in general, they definitely need to substantiate it on more datasets.

Overall I think the experimental setup and the writing of the paper needs improvements.

---

> ### Author Response · Authors · 2023-04-04
> **Follow-up question**
>
> Thanks for the helpful comments. Adding additional experiments/datasets now.
>
> A couple questions re: 3 and 4 - in the transfer learning setup, we're concerned with contributions of the source data and transfer technique to improve a baseline model which has a known amount of starting information about the target task prior to transfer? A common setting is low starting information/inductive biases which is assayed by low performance of the baselines on the target task prior to transfer (note the conversation with reviewer 13yt). We provide a baseline for this low-information setup, as well as for a common transfer technique - few-shot prompting. Re: regimes with more prior information, anecdotally, we also observed similar gains using Instruct-GPT as both teacher and student model, but did not include these for reproducibility (will include a larger model experiment in the updated revision).

---

### Decision · Action_Editors · 2023-05-22

**Recommendation:** Reject

**Comment:**

The paper considers the challenging task of transfer learning and proposes an interesting method, but the reviewers unanimously found the writing, choice of experiments and empirical results to be lacking in several respects. The authors' response was not sufficient to address the above mentioned concerns, and the paper is unfortunately not up to publishable standards for TMLR.

**Audience:**

The ML sub-community working on transfer learning would be interested in some ideas in this paper, albeit if present more clearly and with stronger empirical studies.

**Claims And Evidence:**

The reviewers found the paper's writing to be sub-par, with a lack of several important details around the method, data collection process, and without rationales for important choices like the evaluation datasets. Reviewers also found the empirical evidence to be lacking, with the paper using weak/insufficient baselines and the method itself not achieving strong performance to back up the claims around it suitability for transfer learning.